# An Adequate Pharmaceutical Quality System for Personalized Preparation

**DOI:** 10.3390/pharmaceutics15030800

**Published:** 2023-03-01

**Authors:** Marta Uriel, Diego Marro, Carlota Gómez Rincón

**Affiliations:** Faculty of Health Sciences, Universidad San Jorge, 50830 Villanueva de Gállego, Spain

**Keywords:** personalized preparation, pharmaceutical quality system, proficiency testing program, magistral formula, official formula, stability, individualized medicine, quality control, pharmaceutical compounding

## Abstract

The pharmacy compounding of personalized preparations has evolved a great deal, and with it, the way of working and the legal requirements have also evolved. An adequate pharmaceutical quality system for personalized preparations presents fundamental differences with respect to the system designed for industrial medicines since the size, complexity, and characteristics of the activity of the manufacturing laboratory and the applications and uses of the manufactured medicines must be taken into account. Legislation must advance and adapt to the needs of personalized preparations, filling the deficiencies currently found in this field. The limitations of personalized preparation in its pharmaceutical quality system are analysed and a method based on a proficiency testing program specially designed to overcome these limitations is proposed: the Personalized Preparation Quality Assurance Program (PACMI). This method makes it possible to expand the samples and destructive tests, and dedicate more resources, facilities, and equipment. It allows for more in-depth knowledge of the product and the processes used, and for proposed improvements that increase the overall quality for improved patient health. PACMI introduces tools used in risk management in order to guarantee the quality of an essentially heterogeneous service: personalized preparation.

## 1. Introduction

The industrialization of medicines was a revolution in therapeutics through its impact on people’s ability to access medications. This industrialization process also engendered the “population dosage” [1], a way of treating pathologies and patients, in addition to limiting the vade mecum to the most frequent pathologies, most common routes of administration and strengths, and most stable pharmaceutical forms. This has been exacerbated by successive economic situations resulting in reductions in certain medicines because they are not profitable enough for the industry. All this underscores the fact that personalized preparations (medicine prepared by a pharmacist for a specific patient in response to a specific medical prescription) continue to be essential in modern therapeutics.

This way of compounding medicines—adapted to the patient and their needs—enables us to cover therapeutic areas that are excluded under industrially manufactured medicines, and to resolve therapeutic situations, backed by a doctor’s prescription, which industrially manufactured medicines cannot. Examples of these situations include:Dosage adjustments;Adaptation of pharmaceutical forms and routes of administration;Adaptation of pharmacy compounding to intolerances and pathologies;No treatment: rare diseases, veterinary medicines, shortages, etc.;Instability of certain medicines;Improved adherence to treatment;Economic upswing in treatment, etc. [2,3].

The pharmacy compounding of personalized preparations has evolved notably over the last few decades. We can now consider it a way of providing pharmaceutical care, offering the patient personalized pharmacotherapy [4], ranging from traditional of magistral formulas and official formulas to pioneering advances such as 3D printing of medicines or CAR-T cell therapy. The way of working and the legal requirements involved have evolved accordingly. Regulations in this ambit aim to ensure that the patient has quality personalized preparations.

Quality is the degree to which a series of properties inherent to a product, system, or process meets the requirements [5]. Quality is essential in both products and services. Quality is mandatory given medicine is a product intended to improve the health of patients. The quality of medicines should ensure their suitability for their intended use, that they comply with the requirements and do not pose a risk to patients, i.e., the medicine is safe, effective, and stable during the period of intended use. This means that a pharmaceutical quality system must be in place in every laboratory that produces and manufactures medicines. Their quality is ensured throughout the production process, from start to finish. Therefore, the quality of a personalized preparation produced under a quality assurance plan would then be ensured in the same way as for industrially manufactured medicines [6].

Quality assurance covers all processes related to the acquisition, reception, cleaning, handling, manufacturing, conditioning, and conservation of the raw materials, tooling, packaging material, and intermediate and finished products. The personnel involved in the processes are responsible for ensuring all these aspects. The way to accomplish this, and in particular, to demonstrate it, is by documenting all aspects related to the preparation of medicines: everything must be written down; the process must be undertaken as written; a record kept of what is done; and every step taken must be checkable. This is called traceability [7].

The pharmaceutical quality system covers the rules for good preparation, as well as quality controls; in general, it includes everything that may impact product quality. In creating the system, the size, complexity, and characteristics of the activity of the manufacturing laboratory must be contemplated. Thus, the pharmaceutical quality system in place for a laboratory that produces within a pharmacy or hospital pharmacy service must be different from the industrial pharmaceutical quality system. For the purposes of this article, they shall be referred to as “compounding laboratories”.

In order to delve into the pharmaceutical quality system for personalized preparations, it is necessary to assess the legislation regulating this practice.

## 2. Personalized Preparation Quality: Definitions and Legislation

There have been innumerable studies into quality management for industrially manufactured medicine production. A slew of legislations, regulations, and guidelines have been pursued by national and international organizations. Two fine examples are the Guidelines for Good Manufacturing Practice for Medicinal Products for Human and Veterinary Use (from the European Commission [8]) and the Good Manufacturing Practices (GMPs), created by the Food and Drug Administration (FDA) and disseminated internationally [9].

The specifications of raw materials, definitions, description of controls, and much more information are collected in the various pharmacopeias—European (E.P.) [10], United States Pharmacopeia (USP), and others—and, likewise, the legislation that regulates them. In the European Pharmacopoeia, there is no specific section on magistral compounding. Nor are there any monographs on the finished product. By contrast, the USP does have specific sections (795 and 797) [11,12]. It contains monographs about compounded preparations with specifications that can be taken as a point of reference.

The regulations, norms, and debate on personalized preparation quality can be considered a field of study that is underdeveloped. Many of the regulations established for industrially manufactured medicines are not directly applicable to personalized preparations. One of the aims of this work is to review and discuss the legislation on personalized preparation quality.

Due to globalization in the manufacture and marketing of industrially manufactured medicines, there is an international harmonization of the regulatory authorities and industry representatives of the largest world powers (the United States, Canada, Europe, and Japan), that aims to square the technical and scientific requirements for the registration and authorization of medicines [13]. As pharmacy compounding for personalized preparations is for a specific patient and pathology, the unification of criteria is not needed in response to commercial criteria; therefore, there has been no harmonization process in this field. However, these are not the only criteria that should be considered. Rather a more relevant common criterion is the accessibility of medicines and the safety of patients around the world.

There are references to the unification of criteria in some regions, such as the proposed Latin American Formulary, fostered by the Latin American Medication Authorities network (EAMI). This is a response to the existing need in most Latin American countries to establish quality criteria for the production of magistral and official formulas. Thus, this organization has articulated a joint document that envisions the minimum requirements to be met by pharmacy offices and hospital pharmacy services producing magistral and official formulas [14].

At a European level, Directive 2001/83/EC [8] defines a magistral formula as a medicine prepared in a pharmacy from a medical prescription intended for a certain patient; and an official formula as medicines prepared in a pharmacy from the indications of a pharmacopeia, intended to be dispensed directly to patients served by said pharmacy. These magistral and official formulas comprise what is referred to in this article as “personalized preparations”, which is a term that is both more current and more precisely defines their use and function. This directive thus exempts personalized preparations from compliance with manufacturing and marketing authorization. The Guide to Good Practices in the preparation of medicinal products in health establishments—created by the Pharmaceutical Inspection Convention (PIC) [15]—is another European publication worthy of note. This guide offers guidance on Good Practices in the preparation of medicines for direct supply to patients that are produced by healthcare establishments. Similarly, Resolution CM/Res (2016) 1 [16] on quality and safety requirements for medicines for the special needs of patients prepared in pharmacy aims to bring about unification in the legislation of all countries. This standard recommends that the GMP Guide be used as a reference for an appropriate quality system for “high-risk preparations”, and that the PIC/S GPP Guide be used for “low-risk preparations”. Analyses of the impact of this resolution on different European countries have been conducted [17,18,19].

The European regulatory framework for personalized preparations (with references also to regulations in the United States) is duly covered in the review by Minguetti et al. [3]. In its conclusions, it proposes, at a European level, to allow pharmacies to prepare medicinal products on behalf of other pharmacies, certify high-risk preparations (e.g., sterile ones), and provide the patient with a leaflet detailing essential information. It further concludes that it is time to modernize how personalized preparations are regulated. It should be noted that everything proposed in this review article is complied with in the Spanish legislative framework: pharmacies can prepare products on behalf of other pharmacies; the needs and requirements to be included in leaflets accompanying magistral and official formulas are included in legislation; and in March 2022 the Spanish Agency for Medicines and Health Products and the Ministry of Health announced—in a public consultation—a proposed Royal Decree (RD) to modify RD175/2001, though no draft is yet available. For this reason, Spanish national legislation is used to frame the regulations governing the compounding of personalized preparations and to enable the subsequent discussion about the requirements of the pharmaceutical quality system, which is the main objective of this article.

The legislation applied in Spain to the quality of personalized preparations can be found in Table 1:

On 22 March 2022, the Spanish Agency for Medicines and Health Products and the Ministry of Health announced, in public consultation, a proposed Royal Decree to modify RD175/2001. No draft is yet available, but it is an opportunity to improve and modernize the legislation. Some of the issues where progress needs to be made are as follows:Allow preparations to be available for stock, so as to respond to the immediate needs of some patients. This already happens in some European countries, per the definition of “stocks preparation” provided by the European pharmacopeia: pharmaceutical preparations prepared in advance and stored until a supply request is received [10].Increase the typified formulas, and do not require these to be included in a formulary, since this system is not dynamic; rather it is based on a scientific justification that demonstrates its effectiveness and safety.Extend the regulations to the development of all routes of administration and pharmaceutical forms, e.g., that sterile forms be included. This will make it possible to cover all prescriptions and needs, guaranteeing the required quality.Address the need to consider the differences between industrially manufactured and personalized medicine.Consider the concept of designed quality (a concept discussed in detail in this article).Consider the demands of the next generation of personalized medicines, such as 3D Printing and CAR-T cell therapy.

## 3. A Review of the Evolution of Quality Management Systems for Medicines

According to Botet [27] the approach toward medicine quality has evolved over time. The system of “analysed quality”, based on the quality control of finished products, evolved into the concept of “manufactured quality”, which is based on quality during production. This led to the establishment of GMP regulations in 1960, and onwards to the main focus nowadays: “designed quality”. This adds specific product considerations, improvement options, validations, risk management, and so forth compared to the previous perspectives, i.e., resulting in a much more complete vision. The different approaches (analysed, elaborated, and designed quality) are not exclusionary but supplementary, adding the various successive perspectives on quality to the previous blueprint.

These perspectives are detailed below, with an analysis of their comparative application to personalized preparation and industrially manufactured medicine environments.

### 3.1. Quality Control

A quality system is based on controlling the end product. This is how product quality began to be assessed. The aim was to limit the arrival of unsafe medicines on the market.

Quality control has certain limitations from a personalized preparations perspective given their intrinsic characteristics.

In many cases, these are destructive controls. The development of personalized preparations requires a single product to be developed for a specific patient and a specific pathology. For example, if 30 capsules are made for one patient, it makes no sense to use 20 of them for analytical checks.

The analytical needs that some quality controls entail pose a further limitation. Highly qualified equipment is required that is rarely found in a compounding laboratory.

Small-scale production—without manufacturing batches—and the variety of compounds made are therefore limitations for some quality controls that cannot be undertaken and would offer valuable information regarding the final quality of the prepared product, and the suitability of the processes used in its manufacture [15].

The controls for typified magistral and official formulas will be those established in the National Formulary [21]. The pharmacy will keep and store in an appropriate place a sample of each prepared batch of the official preparations for up to one year after the expiration date, to be of sufficient size to allow a complete examination. The only quality control required for all magistral formulas is an assessment of their organoleptic characteristics [23].

### 3.2. Elaborated Quality

An approach based on good manufacturing. The premise that quality does not occur in the control, but in production, has begun to take shape, since the controls ensure faults in product quality are detected. However, unlike the application of standards for good manufacturing practices, they do not help to minimize these faults. Both international (GMP), and national regulations Normas de Correcta Fabricación (NCF) in the case of Spain, arise from the goal of regulating all production-based aspects that may affect the quality of the end product throughout the life cycle of the medicine. The main limitation of these regulations is that they are very generic and not specifically related to a particular product. Thus, the concept of product-oriented validation began to be employed, based on repeating production per an agreed procedure with a final assessment of compliance with the specifications. If the process is successful, then when it is applied to subsequent productions it can be deduced that there will be conformity in the final product [27].

There are many differences in the concept and production of industrial and magistral medicines, meaning the application of the rules for proper production must be different in each case, as detailed below in Table 2:

### 3.3. Designed Quality

Although the preceding approaches contribute to the quality of the medicine, they have certain shortcomings that were addressed by the FDA initiative “GMPs for the 21st Century” (CGMPs). In particular, the following concepts are added to the previous ideas:consideration of the product in particular;options for improvement;validation through variables;risk systems;improved understanding of the processes.

This speaks to a “designed quality”, the basis of which is knowledge of the product. This approach gained worldwide acceptance when it was adopted by the International Council for Harmonisation (ICH), becoming a new paradigm for managing the quality of medicines [32].

## 4. ICH Objectives for a Pharmaceutical Quality System

The “pharmaceutical quality system” as defined in ICH10 [33] describes a comprehensive and effective model based on the quality concepts of the International Organization for Standardization (ISO) [34]. It includes Good Manufacturing Practice (GMP) regulations [9] and other ICH norms. The implementation of a quality management system in the production of medicines would entail achieving three objectives:Making the product using a system that enables the provision of a product of suitable quality for the patient, healthcare professionals, and authorities.Effectively controlling and monitoring processes and product quality and ensuring ongoing suitability. Quality risk management is useful to identify them.Continuous improvement in the quality of the product, processes, and pharmaceutical quality system. Reduce variability. To meet this objective, quality risk management is useful to identify and prioritise areas.

Next, a comparison is undertaken of the application of these objectives to industrial and personalized preparations.

### 4.1. Application of the Objectives to Industrially Manufactured Medicine

The industrial-level pharmaceutical quality system must guarantee compliance with the preceding three objectives. This must occur during the design, development, quality controls, validations, and revalidations of the entire life cycle of the preparation, overseen by the quality manager. There are sufficient resources and no time limitations, since this process is prospective, concurrent, and retrospective, allowing for optimum results in the quality of the manufactured products. One of the main objectives is to ensure that all the units produced are the same, i.e., homogeneity in product and processes.

### 4.2. Application of the Objectives to Personalized Preparations

When reviewing the possibility of compliance with the quality objectives set by ICH in regard to personalized preparations, it can be seen that:The main guarantees for fulfilling objective one in the case of a personalized preparation are scientific knowledge and good practices of the prescribing professional and the pharmacist who makes and dispenses the preparation. These professionals take full responsibility for the quality of the preparation process and the use of the medicine [35].The quality management system applied in the compounding laboratories from the acquisition of the initial materials to the controls and the dispensing of the product ensures that products with the right quality are made, as described in the legislation. In Spain, RD 175/2001 brought about a change in this regard, since it regulates the rules of proper preparation and quality control of magistral formulas and official formulas. In fact, customer satisfaction with personalized preparations is high [36,37,38]. In addition, satisfaction is very high since this is a medication that is designed, prescribed, and developed for a specific patient to treat a particular pathology. These reasons also make it possible to reduce side effects [39]. Compliance with the first objective is thus ensured.Meanwhile, from a quality control perspective, an intrinsic limitation based on the very personalized nature of the preparation has already been described in previous sections: the impossibility of measuring certain attributes that define quality.The second objective is partly fulfilled in the case of personalized preparations since some controls can be instituted that offer information about the process and the quality of the product, though there are certain limitations. Process monitoring strategies can be established using protocols that are the result of compound production validations and detailed knowledge of manufacturing operations. There are restrictions at this point: unitary and heterogeneous production means that the possibility of monitoring by means of certain quality controls is reduced.Quality risk management will make it possible to identify what is important and what is not, meaning that resources can be usefully applied, thereby obtaining the best possible results. The products and their processes must be fully known and understood and must be analysed to identify any real hazards and risks. Once the risk is known, it should be assessed and accepted or rejected depending on its level and the ways in which the risk may be reduced or avoided [27].Detailed knowledge of the product and process management of a personalized preparation is sometimes limited by the resources available for its development and restricted by the variability and time limitations noted elsewhere. Even so, as Minguetti states [3], the authorities agree that the risk associated with the development of personalized preparations is considered acceptable given their added value, even if they are not prepared under the same quality system as at the industrial level.In terms of complying with the third objective, when improvements are identified during the magistral formulas compounding process, they are implemented and incorporated into the work protocols. Restrictions are again observed in the detection of such improvements in relation to the previously mentioned peculiarities of personalized preparations. The limitation of not being able to perform certain controls that provide information about processes can be restrictive when identifying areas for improvement. A need for improvement may be found, though the exact processes affected might not be identified if they are not all monitored.

## 5. Proposal of a Strategy for Overall Improvement in the Personalized Preparation Quality System: Proficiency Testing

This article has thus far shown that there are certain shortcomings in the quality system for personalized preparations. These are summarised as follows:Sampling is impossible as the preparations are single productions (no batches).Destructive controls cannot be performed for the same reason as above.There is limited availability of the necessary equipment for monitoring and quality control.The great variety of medicines produced leads to an increased number of processes and products to be analysed, understood, and mastered.Limited resources (financial and time) are available to design and develop the compounding.A lack of sampling and analytical control equipment leads to limitations in continuous monitoring; thus, improvement needs may be identified, though the processes or operations that are affected may remain unknown.The percentage of error committed rises when handling small quantities of raw material in the processes, e.g., in weighing, mixing, conditioning, etc. [40].The stability studies performed on personalized preparations remain suboptimal.Detection of contamination from previous products is a critical issue. Cleaning validation programs form a key component of a quality system.

Some of these shortcomings or other errors made due to a poorly implemented quality system for personalized medicines have led to the generation of errors that have resulted in harm to the patient [41,42].

The great interest that compounding pharmacists (responsible for the quality of personalized preparations) have in achieving a suitable pharmaceutical quality system justifies the need to promote an overarching improvement strategy in this area [43]. The objective of this strategy would be to resolve the deficiencies found. The key points of the strategy would be as follows:Increase the number of samples and thereby ensure sampling and the possibility of destruction.Have the infrastructure, equipment, and resources that allow increased knowledge of both products and processes.Hence, it is necessary to have a system that allows the maximum number of variables in these products and processes to be compared.

All of this gels perfectly with the design of proficiency or inter-laboratory testing programs; proficiency testing programs successfully used in other areas include biological analysis laboratories and laboratory diagnostics, radio-toxicology [44,45], laboratory or analytical medicine [46,47,48,49], and therapeutic drug monitoring [50,51,52].

The objective of these proficiency testing programs in general is to increase the analysis and knowledge of products to thus contribute to an overall improvement in product quality. The Personalized Preparation Quality Assurance Program (PACMI) is proposed to achieve this objective for the development of personalized preparations.

PACMI was created in 2010 as a transfer service at the Faculty of Health Sciences of San Jorge University. The objective behind PACMI is to fill the gaps in the personalized preparation quality systems. The Good Pharmacy Preparation Practices Program (PBPPF) from the Official College of Pharmacists of the Province of Buenos Aires was taken as an initial point of reference. The work methodology is based on undertaking a comparative analysis of the products made by the compounding laboratories that adhered to the program. PACMI selects a drug on a quarterly basis and requests the compounding laboratories that adhered to the program send samples of it. Once the samples are received, PACMI performs a detailed study that includes mandatory and non-mandatory quality controls, in addition to analysing any other factor that could affect the quality. For example, if a transdermal cream is requested, additional tests are performed in addition to the mandatory tests (organoleptic characteristics), usually collected in the pharmacopeias. The information on the quality of the product is thus completed with aspects including extensibility, microbiological quality, viscosity, active ingredient concentration, pH, and emulsion marks. The pertinence of the documentation used in its preparation is also controlled including the leaflet, labelling, and preparation guide. Additionally, other aspects that may affect the quality of the end product or the manufacturing processes used are evaluated. These include scales used for the weighing and their sensitivity, dosage calculations, supplier and batches of active ingredients, conditioning, and even the assigned price.

Figure 1 shows a graphic summary of the work methodology used by PACMI for each proficiency test.

The objective of all this is the in-depth knowledge of the product and the processes used when producing the personalized preparation. This leads to both an analysis of its quality results and a more in-depth study undertaken by each compounding pharmacist, offering improvement strategies and a risk analysis for the product. The Corrective and Preventive Action (CAPA) system is used for this [33]. The results obtained are included in two documents which are sent to all participants in the test:−An individual report with details of the results specific to each participant. The deviations or non-conformities found are detailed and preventive or corrective measures are proposed. The aim is to input improvements into the processes at the compounding laboratory.−A general report which is duly encoded to safeguard the anonymity of all participants. This contains the comparative results of all the laboratories for all the tests done and a general overview of the incidents or deviations identified related to quality, anonymously identifying the affected samples. These are documented and their root cause is sought. Lastly, corrective or preventive measures are proposed.

The participating laboratories record the proposed measures and monitor the incorporation of these measures into their processes. Thus, the impact of PACMI on quality does not end in the correction of a specific compounding, but will be applied to the continuous improvement of the processes at the compounding laboratory. The successive comparative rounds of PACMI allow the practical effectiveness of the corrective measures adopted to be verified.

The participating compounding laboratories apply the CAPA system contemplating the frequency or likelihood of the detected risk happening, the severity or impact of this risk, and whether it may affect the health of patients. For example, a field missing in the documentation included in the manufacturing guide is not the same as a preparation that does not comply with the content uniformity test.

Therefore, PACMI would cover the deficiencies highlighted above for personalized preparation: it increases the number of samples, makes it possible to have more resources, equipment, and facilities dedicated to quality assurance tests, etc. All this will have a positive impact on the quality of the personalized preparation to the benefit of patient health and the laboratory itself, which may even see a reduction in costs associated with “non-quality”. It shows a commitment to continuous improvement, enabling confidence in the personalized preparation to rise among health authorities and patients.

The conclusions obtained from each study, plus the accumulated experience from the 850 samples analysed across 26 proficiency tests, allow us to build another tool that contributes to organizing all the variables that affect critical quality attributes of personalized preparation formulation: the “fishbone” or Ishikawa Diagram [32,33]. This provides valuable information to identify the root cause when specifications are not met.

Other PACMI results can be highlighted. For example, the number of compounding laboratories has increased in 12 years from 22 participants in 2010 to 83 in 2022, located throughout the national territory. A quality adequacy analysis is performed on the last 420 samples tested in the last 10 rounds. To evaluate quality, controls are used that are carried out in all the rounds and for which an objective assessment of compliance can be established, since they are established in the legislation or in the reference pharmacopeias. The following are included in the analysis: 1. the adequacy of the documentation sent: preparation guide, labeling, and leaflet, 2. microbiological quality, 3. extractable volume, and 4. quantification of the amount of active ingredient compared to the declared amount. Total compliance, which reflects the quality of the formulations analysed, is 90.72% with a standard deviation of 6.4.

A wide variety of compounding laboratories are involved in PACMI. The differences found within the compounding of each personalized preparation in each laboratory make it difficult to identify the root cause of certain incidents. Ishikawa diagrams make this task easier.

An example of a fishbone/Ishikawa diagram is shown in Figure 2.

Once the causes have been identified, PACMI exposes them in general and individual reports and offers solutions. In addition, recommendation guides (informative documents) are created and sent together with the information on inter-comparative circuits. These recommendation guides are posted on the PACMI website, as a reinforcement message for the participating laboratories.

Therefore, PACMI makes a decisive contribution so that personalized preparations can comply with the provisions of ICH10 [33] by ensuring the continuous capacity of processes and controls to make a product of the desired quality and identify areas for continuous improvement.

Other objectives of PACMI include:Gathering information to demonstrate to third parties the effective overall quality of personalized preparations.Undertaking stability studies and galenic developments.

Another fundamental strategy for continuous improvement is the specialized training of compounding pharmacists. This need was detected at San Jorge University, home to PACMI, and postgraduate training was created in the shape of expert degrees, which today constitute a university master’s degree. As of now, some 318 students have taken this degree over the seven years it has been running.

## 6. Quality = Homogeneity/Heterogeneity

Industrially manufactured medicine and personalized preparations present substantial differences throughout their life cycle, which means that their pharmaceutical quality system cannot be the same.

Industrially manufactured medicine needs to duplicate exact copies of a product that meets a series of quality parameters, such that the concept of industrial quality can be equated to that of homogeneity, both in processes and in the end product.

The main difference found between industrially-manufactured and magistral medicines in the description of the ICH preparation objectives (see above) is the association of the idea of homogeneity as a sign of quality, described literally as “the reduction of variability” in objective three. An industrially manufactured medicine is designed and controlled throughout its life cycle, and the definition of quality production for these medicines is associated with the idea that millions of “identical copies” of the design are produced in the manufacturing laboratory. Both the manufacturing processes and the quality controls performed on these medicines aim to demonstrate their homogeneity and consider any deviation from the standard pattern as non-compliant.

In laboratories where personalized preparations are produced, the compounding pharmacist must deal with the design and life cycle of a multitude of different products—as many, in fact, as there are patients and pathologies. Hence, it is not feasible to work with the concept of quality = homogeneity, rather it must be shown that quality can coexist with a certain degree of heterogeneity. This entails design, management, and quality control systems of its own that are different from that of industrially manufactured medicines. The heterogeneity of a personalized preparation is an intrinsic quality of that preparation and does not imply a lack of quality. If a medicine is not standardized, it does not mean that the product in question is not a quality product. It means that we must guarantee its quality through a management system appropriate to this heterogeneity, i.e., through mastery and control of the processes used in its manufacture.

The compounding of personalized preparations should be governed by quality by design, as per ICH8 [53]. This would allow regulatory authorities to consider modifications within the design space as changes that have no impact on the degree of quality of the products compounded. These design modifications make it possible to achieve the objective of personalized preparations, i.e., adapting to a patient and a specific pathology without affecting medicine quality, and substantiating the idea of quality in heterogeneity. It would be fitting for this approach to be taken into account in the latest version of RD 175/2001 that is currently being drafted.

To this effect, the Quality Tested Product Profile (QTPP) is established within each test undertaken by PACMI, defining dosage form, route of administration, dosage and characteristics of APIs, conditioning, and quality criteria.

The main objective of the PACMI study within each proficiency test is to identify critical variables, Critical Quality Attributes (CQA), and Critical Process Parameters (CPP). In addition to identifying them, their acceptable ranges should be determined.

Let us see some examples for the purposes of clarity. The critical quality attribute (CQA) of a personalized preparation containing very little active ingredient would mean that that amount of active ingredient is within concentration limits in the preparation of ±10%, or within the range of 90.0–110% of its declared theoretical concentration. One critical process parameter (CPP) could be the variability of factors that may affect the weighing process and have an impact on the defined CQA. Continuing with this example, the error made in the weighing process can be monitored (it may be due to multiple causes: adequate sensitivity for the weight, calibration, etc.) and the error limits assumed during this process quantified to see what impact these CPP have on the proposed CQA. Naturally, one must do more than merely identify these factors, but all this is undertaken with the final objective of creating a process monitoring/control strategy that ensures the finished product meets the established criteria (QTPP). All these analyses can be performed variable by variable in a unitary way or by trying to combine the effects that some variables have on others, i.e., multivariate analysis.

## 7. Conclusions

Specificities mean a pharmaceutical quality system suitable for personalized preparations presents fundamental differences with respect to a quality system designed for industrially manufactured medicines.

Legislation must advance and adapt to the needs of personalized preparations, filling the deficiencies currently found in this field: unregulated routes of administration and pharmaceutical forms (sterile), the possibility of formulation of products for stock, updating specifications of finished products, etc.

The limitations of personalized preparations in their pharmaceutical quality system are analysed and the Personalized Preparation Quality Assurance Program, PACMI, is proposed to resolve them by:Allowing destructive testing.Increasing the number of samples that facilitate comparative analysis and obtaining conclusions about the impact of some particular factors on the processes.Enabling the dedication of more resources: equipment and facilities such as pharmacy technical equipment, sterile environments for microbiological controls, analytical tools, etc.Expanding knowledge of the product and processes used in the production of a personalized preparation. In addition to analysing specific quality results, it is possible to broaden the study undertaken by each compounding pharmacist and offer improvement strategies and a risk analysis for the product.Pinpointing the deviations or non-conformities found, documenting and identifying their root cause (with the Ishikawa diagram), and proposing preventive or corrective measures (CAPA system). The aim is to input improvements into the processes at the compounding laboratories.Contributing to the overall improvement of product quality through increased analysis and knowledge to the benefit of patient health.Making galenic developments and stability studies.

Products and processes should continue to be analysed in the pursuit of increased knowledge and with it the quality of the personalized preparations compounding.

As used in PACMI, the multivariate analysis manages to establish a design space, a fundamental tool to guarantee the quality of personalized preparations, confirming that heterogeneous products can also be quality products.

## Figures and Tables

**Figure 1 pharmaceutics-15-00800-f001:**
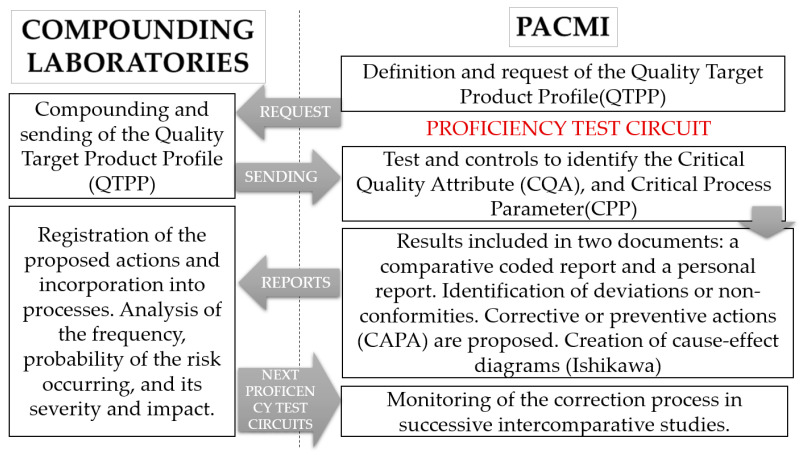
Work methodology in the PACMI proficiency test programs.

**Figure 2 pharmaceutics-15-00800-f002:**
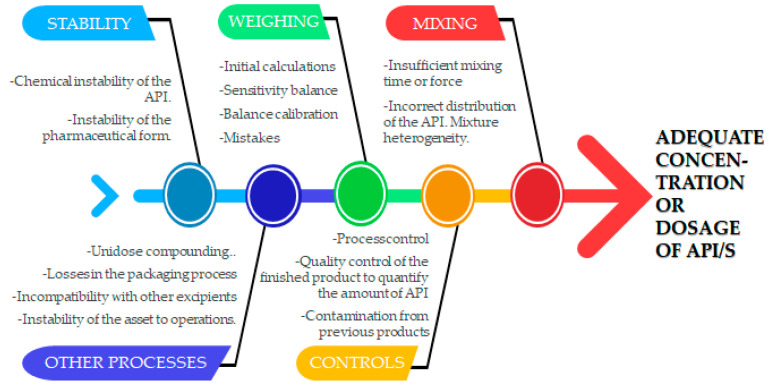
Ishikawa diagram with the categorization of the possible root causes that contribute to noncompliance with specifications in the adequate concentration or dosage of active pharmaceutical ingredients (APIs).

**Table 1 pharmaceutics-15-00800-t001:** Legislation regulating the activity of compounding personalized preparations in Spain.

Law/Norm	What It Regulates	Observations
Royal Legislative Decree 1/2015, of 24 July [20]:	Passes the text of the Law on guarantees and rational use of medicines and health products.	It includes a section that states that magistral formulas and official formulas are legally recognized medicines, prepared by or under the direction of a pharmacist, and dispensed from a pharmacy or pharmaceutical service, complete with sufficient information to guarantee they are correctly identified (including the name of the pharmacist who has prepared them), conserved and safely used.
Royal Decree (RD) 175/2001, of 23 February [21]:	Passes the norms for proper preparation and quality control of magistral formulas and official formulas.	This is a turning point in legislation on personalized preparations. It introduces regulations for proper production and quality assurance applied to compounding laboratories. They are considered the minimum that every compounding laboratory must meet to ensure the quality of the medicines it manufactures. There are certain “gaps” in them which are detailed in this article.
Royal Decree 905/2003, of 11 July [22]:	Modifies the sole transitional provision of Royal Decree 175/2001.	It was published to extend the deadline for the adaptation of laboratories to the requirements of RD175/2001.
Order SCO/3262/2003; updates: SCO/3123/2006 and SSI/23/2015 [23]	The National Formulary (and its subsequent updates) and the Royal Spanish Pharmacopoeia are passed.	Currently, 3rd edition of the formulary. It contains 81 monographs on raw material, 57 on official formulas, and 29 typified magistral formulas: standardized compounding approved by the Formulary and Pharmacopoeia Commission (collected in the National Formulary).
Royal Decree 1015/2009, of 19 June [24]:	Regulates the availability of medicines in special situations and what can be produced.	There are circumstances in which the clinical data that support a certain therapeutic use for an already authorized drug are not included as an officially approved indication. Very relevant in areas with intense research activity and the pace of evolving scientific knowledge that precedes the procedures to incorporate them as an official indication. There are also conditions of use in clinical practice that are not contemplated in authorization for the medicine, given an absence of commercial interest for undertaking studies pursuant to its authorization. These exceptional uses lie within the wardship of clinical practice and are the responsibility of the prescriber. Authorization is not required on a case-by-case basis.
Royal Decree 824/2010, of 25 June [25]:	Pharmaceutical laboratories, manufacturers of active ingredients for pharmaceutical use, and foreign trade in medicinal products and investigational medicinal products	Point 3 of Article 1: non-mandatory authorization as manufacturers of medicines for compounding services that produce magistral and official formulas.
Guide to good medicinal product preparation practices in hospital pharmacy services [26]:	Points not observed in RD 175/2001, which are very necessary for hospital pharmacy services: handling, fractionation, and personalized dosage. It covers the preparation of sterile products.	It arose as a result of Royal Decree-Law 16/2012 on urgent measures to ensure the sustainability of the National Health System and improve the quality and safety of its services, which determines the possibility for the Autonomous Regions to certify hospitals for the tasks of handling, fractionation, and personalized dosage, and to guarantee application of the technical guidelines on good practices in hospital pharmacy services in the absence of said text. As it is not legislative in character, merely a recommendation, it is not mandatory to comply with, rather it must be observed. Application and direct reference to the hospital environment, with the same absences in the same spheres, on the matter of compounding in the regional pharmacy.

**Table 2 pharmaceutics-15-00800-t002:** Differential aspects of industrially manufactured medicines and personalized preparations.

	Industrially Manufactured Medicines	Personalized Preparations
Batch size	Large batches	Mostly as a unit product (a medicine/prescription). Small batches, only in official formulas.
Variability in product/laboratory	A limited number of different medicines/items in each laboratory. Patients adapt to the medicine. The homogeneity of all medicines with the same item number is guaranteed.	Great diversity in medicines; heterogeneity. As many prescriptions as there are patients: subject to changes in pharmaceutical forms and active ingredients; combinations and modification of dosage, excipients, packaging, etc. Great versatility and extensive knowledge are required to be able to pursue development and compounding in a limited timeframe. The medicine is tailored to the patient [16].
Investment of resources in each medicine developed	Each development entails validations and controls, and significant investment of time, money, and personnel. The mean time invested to receive authorization to market a medicine is about 10-12 years with a financial outlay of approximately 800 million Euros [28]. After scaling, production yield is high.	Limited resources and minimal time (from prescription to dispensing: hours or days) are available to design and develop a wide range of medicines. Without these time and resource limitations, it would be unfeasible to be able to respond to the personalized creation of such products at a price that was not exorbitant. As stated in the European resolution on quality assurance and safety requirements for medicines prepared in pharmacies (personalized preparations), the risks of a delay in the supply of a medicine to treat the patient must be assessed, albeit without ignoring the possible risk that could result from an error in preparation. The production is at times singular, making sampling impossible and quality control destructive.
Supplies/quantities of raw materials and medicines	The portions of initial, intermediate, and final products are large and it is produced in batches. The yield achieved with high-quantity production makes it possible to invest effort in continuous improvement and process knowledge.	Very small quantities are handled, both in terms of starting materials, in the quantities weighed, and in the number of preparations that are repeated after each design. Handling small quantities involves difficulties. For example, weighing is common to all manufacturing processes, but weighing small quantities generates greater error. (There is a link between the weighed quantity, the sensitivity of the scales, and the error made in this basic operation).
Profitability	The initial investment of time and resources to develop a new medicine is high, but once it moves to an industrial scale, the investment is profitable through the high yields achieved with industrial equipment and processes.	For each prescription: development includes a search for information, design, development, specific documentation, purchase of materials from suitable suppliers, validation by creating a production protocol, and lastly, preparation of the final compounding. This is often done for a once-off prescription. As for raw materials acquisition in cases of single preparations, it should be discarded without being used because suppliers do not have small enough sizes.
Controls	Production in large batches allows ongoing control. Sampling is not limited, even if the controls performed are “destructive”. Checks can be performed throughout the life cycle of the product.	The total amount produced does not allow for sampling or for destructive quality controls to be performed.
Validation and continuous improvement	Time and effort are invested in validation and continuous improvement processes. The opportunity exists to know the details of each process, including personnel and specialized equipment.	A lack of resources is detected that enables increased knowledge regarding some products made at the magistral level
Stability	All industrially manufactured medicines have full stability studies. The objective is to know the validity period of use for the medicine. They allow an experimental expiration date to be given, which is obtained by means of testing forced degradation, and stability studies at certain times contiguously when marketing the approved batches. Optimum conservation conditions (conditioning, temperature, and humidity) are assessed [29].	Not enough information is available on stability since any alteration made to the prescription (change of dose, excipient, or packaging material) will modify the expiration date of the medicine. These changes are at the heart of personalization. Magistral production does not require preparations with such long-life cycles as they are not manufactured, transported, and stored because they are dispensed shortly (hours or days) after they are produced. Personalized preparations are not “in stock”; this only occurs with official formulas. That is why the term extemporaneous preparations is also used to refer to personalized preparations. The Pharmaceutical Inspection Convention (PIC) guide [15] defines an extemporaneous preparation as a product that is dispensed immediately after preparation and not kept in stock. Legislation limits the expiration date of magistral formulas (those that are not typified) to the duration of the prescribed treatment [21]. There is a lack of information when assigning expiration dates to personalized preparations and estimates or decision algorithms must sometimes be used. There are resources that help with this task, such as the manual “Trissel’s stability of compounded formulations” [30] and the “Stabilis” hospital pharmacy website: http://www.stabilis.org/ (accessed on 5 December 2022) [31]. There are pharmacopeias and manuals that compile decision algorithms for the assignment of expiration dates when there is no bibliography, such as United States Pharmacopeia (USP) or the guide for good practices for the preparation of medicines in hospital pharmacy services [26].
Legislation (Spain is used as in the previous section)	Sections that contain Spanish national legislation on the correct preparation and control of medicines at an industrial level, detailed in Royal Decree 824/2010 [25]: Section 1: Pharmaceutical quality system Section 2: Personnel Section 3: Premises and equipment Section 4: Documentation Section 5: Production Section 6: Quality control Section 7: Subcontracted activities Section 8: Complaints, quality defects, and product recalls Section 9: Internal inspections	Sections that contain Spanish national legislation on the correct preparation and control of personalized preparations, detailed in Royal Decree 175/2001 [25]: Section 1: Personnel Section 2: Premises and tools Section 3: Documentation Section 4: Raw materials and packaging material Section 5: Production Section 6: Dispensing Table 1 offers an overview of all the legislation regulating the compounding of personalized preparations in Spain

## Data Availability

Not applicable.

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
