# Peer review of "An Adequate Pharmaceutical Quality System for Personalized Preparation"

_pharmaceutics, 2023, doi:10.3390/pharmaceutics15030800_

Round 1
Reviewer 1 Report
Suggestion that title of "An adequate pharmaceutical quality system for the personalized preparation" would be better English.
Please see file for other recommendations.
I think abstract could be rewritten to be in an abstract format.
There is a lot of unsubstantiated lines/statements/paragraphs in the into and throughout paper. While I do not expect citations everywhere -can the authors ensure they dot make statements that they can't back up.

Author Response
Response to Reviewer 1 Comments
Point 1: Suggestion that title of "An adequate pharmaceutical quality system for the personalized preparation" would be better English.
Response 1: We agree with the reviewer. We have already changed the title as you can verify.
Point 2: Please see file for other recommendations.
Response 2: Thank you, we have made the changes in the recommendations indicated. From line 85 to 86 and from line 103 to 104. The abstract has also been modified, as stated in the following comment
Point 3: I think abstract could be rewritten to be in an abstract format.
Response 3: We agree with the reviewer. The abstract has been modified: lines 10-12 and 16-17.
Point 4: There is a lot of unsubstantiated lines/statements/paragraphs in the into and throughout paper. While I do not expect citations everywhere -can the authors ensure they dot make statements that they can't back up.
Response 4: We agree with the reviewer. We have added some new reference that supports the statements. References:35 and 43. Also, some comments have been removed: lines: 85, 103, 455-460 and 508-512
Ceppi, H.J. Guía práctica sobre recetas magistrales para odontología y otras consideraciones concomitantes. BSG Basago Libros: Córdoba, Argentina, 2003.
Line 278
Puerto Cano, R. Evolución del medicamento individualizado en España, situación actual y perspectivas de futuro. Academia de Farmacia Santa María de España de la Región de Murcia: Cartagena, España, 2021; pp. 99-110.
Line 343

Reviewer 2 Report
Line 38 compounding medicines…..
96 and monographs about compounded preparations
105 Add Canada
Table 155 -156 /224-225 needs to be formatted for better readability
413 APIs
434-439 this makes no sense. Compounded preparations must follow a compounding monograph or master compounding record. Deviations must be documented because this can cause changes in stability or performance! The paragraph sounds like every preparation can be different even if the same prescription was compounded. This needs to be clarified.
The initial mater compounding record was created to address an problem or treatment only few patients have and the dose or amount 50 g 100 g are variable – the rest must be constant.
481 USP 0.10% error!! See USP 41
487 a design space is defined by a statistical design; this goes far beyond what compounding was designed for and its purpose.
The proposed PACMI is probably a model for commercial compounding but not for normal compounding pharmacies. The cost would be too high and it would kill compounding! This is what happened in Canada when USP 800 based requirements were implemented.
The authors should look into the NAPRA guidelines in Canada (free access)
Here they can find some good tables and competencies required for compounding.
In Canada/US excipient companies publish master formulation records and they perform stability testing data. Pharmacies following the monograph are not required to perform stability testing as long as they follow the monograph.
Author Response
Response to Reviewer 2 Comments
Point 1: Line 38 compounding medicines…..
Response 1: We thank the reviewer for this comment . We have modified preparing by compoundig medicines. Line 40
Point 2: 96 and monographs about compounded preparations
Response 2: We thank the reviewer for this comment . We have modified finished product monographs by monographs about compounded preparations. Line 100-101
Point 3: 105 Add Canada
Response 3: We apologize for forgotting to include Canada. We have added it. Line 110
Point 4: Table 155 -156 /224-225 needs to be formatted for better readability
Response 4: We agree with the reviewer. Tables 1(Line 161) and 2(line 234) have been formatted for better readability.
Point 5: 413 APIs
Response 5: We thank the reviewer for this comment . We have modified API by APIs. Line 440
Point 6: 434-439 this makes no sense. Compounded preparations must follow a compounding monograph or master compounding record. Deviations must be documented because this can cause changes in stability or performance! The paragraph sounds like every preparation can be different even if the same prescription was compounded. This needs to be clarified.
The initial mater compounding record was created to address an problem or treatment only few patients have and the dose or amount 50 g 100 g are variable – the rest must be constant.
Response 6: We appreciate the reviewer's comment. It was intended to express the diversity of master compounding record that can be compound in personalized preparation, but, the paragraph is misleading and has therefore been removed. Line 466-471
Point 7: 481 USP 0.10% error!! See USP 41
Response 7:We have reviewed the USP 41 again and found no differences. I quote what USP collects:
“The purpose of strength, or potency, testing is to establish or verify the concentration (strength, potency) of the API in the compounded preparation. USP has established that the aceptable range of most compounded preparations is typically ±10%, or within the range of 90.0%–110.0%.”
Perhaps it was not well understood by the way of expressing it. For this reason, the sentence has been modified, using the terminology cited in USP:
"Let us see some examples for the purposes of clarity. The critical quality attribute (CQA) of a personalized preparation containing very little active ingredient would mean that that amount of active ingredient is within concentration limits in the preparation of ±10%, or within the range of 90.0%–110% of its declared theoretical concentration.
Line 508-509
Point 8: 487 a design space is defined by a statistical design; this goes far beyond what compounding was designed for and its purpose.
Response 8: We agree with the reviewer. Design space is something much more complex, so that statement has been removed. Line 521-523
Point 9: The proposed PACMI is probably a model for commercial compounding but not for normal compounding pharmacies. The cost would be too high and it would kill compounding! This is what happened in Canada when USP 800 based requirements were implemented.
Response 9: PACMI is a program that has been working for 12 years now. The cost per analysis is €100 per sample since intercomparative studies are carried out on a high number of samples. The price of the service has been agreed with the pharmacies and is affordable.
Point 10: The authors should look into the NAPRA guidelines in Canada (free access)
Here they can find some good tables and competencies required for compounding.
Response 10: We thank the reviewer for the recommendation. We have downloaded all the NAPRA guidelines and I find them very useful. I will share them with other colleagues
Point 11: In Canada/US excipient companies publish master formulation records and they perform stability testing data. Pharmacies following the monograph are not required to perform stability testing as long as they follow the monograph.
Response 11: We agree with the reviewer. We are aware of this practice, we have participated in some studies such as these:
Dijkers, E., Nanhekhan, V., Thorissen, A., Marro, D., & Uriel, M. (2017). Limited influence of excipients in extemporaneous compounded suspensions. Hospital Pharmacy, 52(6), 428-432.
Uriel, M., Gómez-Rincón, C., & Marro, D. (2018). Stability of regularly prescribed oral liquids formulated with SyrSpend® SF. Die Pharmazie-An International Journal of Pharmaceutical Sciences, 73(4), 196-201.

Reviewer 3 Report
Any review of personalized medicine must not only look at current practices in personalized medicine/ compounding but also new developments or potential in the area. The reason for this is that any proposed quality system should be capable of meeting the demands of the next generation of personalized medicines without the need for an overhaul of the proposed system. I would like to see at least some reference to the implication of 3D Printing and CAR-T cell therapy on any proposed quality system.
The authors provide a single sided view towards the benefits of pharmaceutical compounding and personalized medicine. A little balance should be provided to highlight the dangers of a poorly implemented quality system for personalized medicinces.
An example from the below paper would be beneficial
Watson CJ, Whitledge JD, Siani AM, Burns MM. Pharmaceutical Compounding: a History, Regulatory Overview, and Systematic Review of Compounding Errors. J Med Toxicol. 2021 Apr;17(2):197-217. doi: 10.1007/s13181-020-00814-3. Epub 2020 Nov 2. PMID: 33140232; PMCID: PMC7605468.
When discussing the role of the ICH, it must be noted that this organisation is not solely represented by the regulatory authorities of the US, Europe and Japan. Industry representatives from the US, (PhRMA) Europe (EFPIA) and Japan (JPMA) were involved from the outset
In Section 5 the authors summarize the shortcomings in a quality system for personalized medicines that can ultimately be addressed by PACMI. There is no mention of contamination from previous products which I imagine to be a critical issue as in "industrial medicine" cleaning validation programmes form a key component of any quality system.
Ultimately the manuscript proposes PACMI as a means to improve the quality system for a personalized medicine but as PACMI was introduced in 2010 I feel the authors need to provide some quantitative evaluation or substantive feedback of its performance to date over 12 years rather than state the number of samples tested and the variety of tests performed. Section 5 also looks at how the use of a Fishbone diagram helped identify the root cause of certain incidents but no information was provided as to how these can be/were addressed by a quality system such as PACMI.
I am not in full agreement with the premise of Section 6 that quality comes down to an interpretation of homogeneity or heterogeneity and that this is the key difference between industrial and personalized medicines. The authors are advocating flexibility in processes used to product a personalized medicine. The choice of words here are open to interpretation. When discussing the quality of a pharmaceutical finished product homogeneity and heterogeneity are often used to describe the uniformity of the finished product particularly in relation to the content of APIs and excipients.
Below are some suggestions for the authors which are solely a personal opinion which can be implemented to improve the readability of the manuscript but do not have to be implemented if the authors do not agree
- Formatting of Tables 1 and 2 could be improved to make it more legible.
- I am not entirely in agreement with the description of commercially produced medicines as "industrial medicines". It is not a description I have seen mentioned elsewhere and does not serve as an antonym to "personalized medicines".
- Figure 2 is graphically (visually) a poor representation of a fishbone diagram.
- To a lesser extent Figure 1 should be revised to improve quality / resolution prior to publication
Author Response
Response to Reviewer 3 Comments
Point 1: Any review of personalized medicine must not only look at current practices in personalized medicine/ compounding but also new developments or potential in the area. The reason for this is that any proposed quality system should be capable of meeting the demands of the next generation of personalized medicines without the need for an overhaul of the proposed system. I would like to see at least some reference to the implication of 3D Printing and CAR-T cell therapy on any proposed quality system.
Response 1: We agree with the reviewer. Some references to new developments in this area have been added:
"ranging from traditional of magistral formulas and official formulas to pioneering ad-vances such as 3D printing of medicines or CAR-T cell therapy." Line 53-55.
"-That the demands of the next generation of personalized medicines be contemplated, such as 3D Printing and CAR-T cell theraphy." Line 181-182
Point 2: The authors provide a single sided view towards the benefits of pharmaceutical compounding and personalized medicine. A little balance should be provided to highlight the dangers of a poorly implemented quality system for personalized medicinces.
An example from the below paper would be beneficial
Watson CJ, Whitledge JD, Siani AM, Burns MM. Pharmaceutical Compounding: a History, Regulatory Overview, and Systematic Review of Compounding Errors. J Med Toxicol. 2021 Apr;17(2):197-217. doi: 10.1007/s13181-020-00814-3. Epub 2020 Nov 2. PMID: 33140232; PMCID: PMC7605468.
Response 2:We agree with reviewer. The vision of the dangers and risks of a poorly implemented quality system for personalized medicines has been included. The reference suggested by the reviewer has been incorporated into the bibliography(41 and 42), added to other that reinforce the idea:
"Some of these shortcomings or other errors made due to a poorly implemented quality system for personalized medicines have led to the generation of errors that have resulted in harm to the patient. "Line 338-340.
References included:
- Watson, C.J.; Whitledge, J.D.; Siani, A.M.; Burns, M.M. Pharmaceutical Compounding: a History, Regulatory Overview, and Systematic Review of Compounding Error. J Med Toxicol. 2021,17 (2), 197-217.
- Staes, C.; Jacobs, J.; Mayer, J.; Allen, J. Description of outbreaks of health-care-associated infections related to compounding pharmacies. Am J Health Syst Pharm. 2013, 70(15), 1301-1312.
Point 3: When discussing the role of the ICH, it must be noted that this organisation is not solely represented by the regulatory authorities of the US, Europe and Japan. Industry representatives from the US, (PhRMA) Europe (EFPIA) and Japan (JPMA) were involved from the outset
Response 3: We thank the reviewer for the recommendation. Industry has been included in the assertion. Line 109-110
Point 4: In Section 5 the authors summarize the shortcomings in a quality system for personalized medicines that can ultimately be addressed by PACMI. There is no mention of contamination from previous products which I imagine to be a critical issue as in "industrial medicine" cleaning validation programmes form a key component of any quality system.
Response 4: We apologize for forgotting to include such an important aspect. It has been included as an item in section 5:
"Detection of contamination from previous products. It is a critical issue. Cleaning validation programmes form a key component of quality system."
Line 336-337
Point 5: Ultimately the manuscript proposes PACMI as a means to improve the quality system for a personalized medicine but as PACMI was introduced in 2010 I feel the authors need to provide some quantitative evaluation or substantive feedback of its performance to date over 12 years rather than state the number of samples tested and the variety of tests performed.
Response 5: The objective of this article was to review the situation of the quality of the current individualized medication and propose PACMI as a support tool for the quality assurance of the individualized medication. That we are finalizing an article with all the quantitative results obtained in PACMI and that we will send it as soon as possible. That is why they had not been included in this article, but in order to respond to the reviewer's request, the following have been included:
“Other PACMI results that can be highlighted: The number of compunding laboratories has increased in these 12 years, going from 22 participants in 2010 to 83 in 2022 located throughout the national territory. A quality adequacy analysis is performed on the last 420 samples tested in the last 10 rounds. To evaluate its quality, controls are used that are carried out in all the rounds and for which an objective assessment of compliance can be established, since they are established in the legislation or in the reference pharmacopoeias. The following are included in the analysis: 1. the adequacy of the documentation sent: preparation guide, labeling and leaflet, 2. microbiological quality, 3. extractable volume and 4. quantification of the amount of active ingredient compared to the declared amount. Total compliance, which reflects the quality of the formulations analyzed, is 90.72% with a standard deviation of 6.4.”
Line 421-431
Point 6: Section 5 also looks at how the use of a Fishbone diagram helped identify the root cause of certain incidents but no information was provided as to how these can be/were addressed by a quality system such as PACMI.
Response 6: We agree with reviewer. It was necessary to explain how PACMI can adressed it. A paragraph has been included to explain it briefly. Line 441-445:
"Once the causes have been identified, PACMI exposes them in the general and individual reports and offers solutions to solve them. In addition, recommendation guides (informative document) are created and are sent together with the information of inter-comparative circuits. These recommendation guides are posted on the PACMI website, as a reinforcement message for the participating laboratories."
Point 7: I am not in full agreement with the premise of Section 6 that quality comes down to an interpretation of homogeneity or heterogeneity and that this is the key difference between industrial and personalized medicines. The authors are advocating flexibility in processes used to product a personalized medicine. The choice of words here are open to interpretation. When discussing the quality of a pharmaceutical finished product homogeneity and heterogeneity are often used to describe the uniformity of the finished product particularly in relation to the content of APIs and excipients.
Response 7:We thank the reviewer for the comment. Significant changes have been made to section 6 in order to clarify this. Line 466-471 and 519-523
Below are some suggestions for the authors which are solely a personal opinion which can be implemented to improve the readability of the manuscript but do not have to be implemented if the authors do not agree
Point 8: Formatting of Tables 1 and 2 could be improved to make it more legible
Response 8:We agree with the reviewer. Tables 1(Line 161) and 2(line 234) have been formatted to make it more legible.
Point 9: I am not entirely in agreement with the description of commercially produced medicines as "industrial medicines". It is not a description I have seen mentioned elsewhere and does not serve as an antonym to "personalized medicines".
Response 9: We agree with the reviewer. We have changed the term “Industrial medicine” to “Industrially manufactured medicines”. It can be more explanatory. Changes are highlighted in blue throughout the article.
Point 10: Figure 2 is graphically (visually) a poor representation of a fishbone diagram
Response 10: Thanks for the comment. The figure has been visually modified. Line 437
Point 11: To a lesser extent Figure 1 should be revised to improve quality / resolution prior to publication
Response 11: We appreciate the comment. the figure has been modified. Line 382

Round 2
Reviewer 3 Report
As the authors have implemented all of the suggestions from my initial review, I feel that the manuscript has been sufficiently improved that I recommend it be accepted for publication.